# Were *Arenga* Palms (Arecaceae) Present in the Eocene? A Review of the Genus *Succinometrioxena* Legalov, 2012

**DOI:** 10.3390/life13051121

**Published:** 2023-05-01

**Authors:** Andrei A. Legalov

**Affiliations:** 1Institute of Systematics and Ecology of Animals, Siberian Branch, Russian Academy of Sciences, Novosibirsk 630091, Russia; fossilweevils@gmail.com; 2Department of Ecology, Biochemistry and Biotechnology, Altai State University, Barnaul 656049, Russia; 3Department of Forestry and Landscape Construction, Tomsk State University, Tomsk 634050, Russia

**Keywords:** Coleoptera, Metrioxenini, *Succinometrioxena andrushchenkoi*, new species, Caryoteae, host plants, North America, Europe, Eocene

## Abstract

It has been suggested that palms of the genus *Arenga* (Arecales: Arecaceae) or forms close to it were distributed in the Eocene of North America and Europe. Records of Metrioxenini (Belidae), which are monophages on these palms, confirm this assumption. A new species, *Succinometrioxena andrushchenkoi* Legalov, sp. n. from Baltic amber is described. The new species differs from *S*. *poinari* Legalov, 2012 in the smaller body sizes, elytral punctation larger than the distances between them, and a rostrum weakly curved in females. It is distinguished from *S*. *bachofeni* Legalov, 2013 and *S. attenuata* Legalov et Poinar, 2020 by the forehead lacking horn-like tubercles on either side of the eyes. A description of male of *S*. *poinari* was herein compiled for the first time. A list and key to fossil Metrioxenini were given. The modern and fossil distribution of the tribe Metrioxenini and Arenga palms was shown.

## 1. Introduction

In the study of fossil fauna and flora, the situation occurs when either plants or insects are poorly represented in the locality. A striking example is late Eocene amber of Europe, where a huge number of inclusions of various insects were found [1], while the plant remains were much rarer and poorly studied [2,3].

Coleoptera are an important component of both modern and fossil ecosystems. Many of these insects have a well-defined food specialization. Herbivorous insects are divided into poly-, oligo-, and monophages [4]. In paleontological reconstructions, it is optimal to use representatives of the last two groups associated in their development with several genera of plants within a family or with several closely related species within a genus.

Obtaining results from Quaternary studies is the simplest study since these are modern species whose trophic relationships are known. The presence of these species in sediments allows us to confirm the presence of its host plants in that locality. For example, finds of *Trichapion simile* (Kirby, 1811) indicate the presence of *Betula*, that of *Phytobius leucogaster* (Marsham, 1802) the presence of *Myriophyllum,* that of *Aizobius sedi* (Germar, 1818) the presence of *Sedum*, and finally that of *Hylobius excavatus* (Laicharting, 1781) and of *Pissodes insignatus* (Boheman, 1843) the presence of *Larix* [5,6,7].

Modern species of beetles, as a rule, are absent in the Paleogene and Neogene. Therefore, the reconstruction of vegetation has to be carried out based on the trophic relationships of weevil genera or tribes, which can also be associated with certain plants. For example, the records of some species of *Ceutorhynchus* Germar, 1823 in Baltic and Rovno amber show the presence of Brassicaceae [8,9,10,11], and *Oxycraspedus* Kuschel, 1955 indicates the presence of *Araucaria* [10,11,12] in the amber-bearing forests.

In the presented study, the possible presence of *Arenga* Labill. ex DC. palms in the Eocene are discussed based on weevils of the tribe Metrioxenini Voss, 1953 which are obligate herbivores of these palms [13]; additionally, a new species of the genus *Succinometrioxena* Legalov, 2012 is described from Baltic amber.

## 2. Materials and Methods

The studied specimens are deposited in Kaliningrad Regional Amber Museum, Kaliningrad, Russia (KRAM); the Institute of Systematics and Ecology of Animals, Siberian Branch, Russian Academy of Sciences, Novosibirsk (ISEA); Harvard University, Museum of Comparative Zoology, Cambridge, MA, USA (MCZ); the Poinar amber collection maintained at Oregon State University, Corvallis, OR, USA (PACO); Andrzej Górski private collection (Bielsko-Biała, Poland) (CAG); Center of Natural History (formerly Geological–Paleontological Institute and Museum) (Hamburg, Germany) (GPIH); Borissiak Paleontological Institute of the Russian Academy of Sciences (Moscow, Russia) (PIN).

Photographs of *Succinometrioxena andrushchenkoi* Legalov, **sp. n.**, *Archimetrioxena zherikhini*, *Succinometrioxena bachofeni*, and *S*. *poinari* were taken using a Zeiss Stemi 2000-C dissecting stereomicroscope, and photos of *Paltorhynchus narwhal* were taken using a Leica M165C binocular microscope. Photographs of *Archimetrioxena electrica* were received from Ulrich Kotthoff and Eva Vinx (Germany: Hamburg).

I used previous reports from the literature [13,14,15,16,17,18,19,20,21,22,23,24,25,26,27,28,29,30,31,32] and collection data to show the recent distributions and localities of fossil forms.

The morphological terminology used in this paper follows Lawrence et al. [33].

Fossil Metrioxenini has been described from the three localities: Roan Mountain—USA: Colorado, Garfield County, Green River Formation; lower Eocene, Ypresian, 50.6–48 Ma; Baltic amber—Russia: Kaliningrad Oblast, Baltic Sea coast and Yantarnyi quarry near Kaliningrad, Poland: Gda’nsk city area, at the Wisla River estuary, Prussian Formation, upper Eocene, Priabonian, 36.8–36.4 Ma; Florissant—USA: Colorado, Rocky Mountains near Pike’s Peak, Florissant Formation; uppermost Eocene, Priabonian, 34.07 ± 0.10 Ma.

Nomenclatural acts introduced in the present work are registered in ZooBank (www.zoobank.org) under LSID urn:lsid:zoobank.org:pub: D03B44CD-D357-4AEE-B982-028E75FE1625

## 3. Results

### Systematic

Family **Belidae** Schoenherr, 1826

Subfamily **Oxycoryninae** Schoenherr, 1840

Tribe **Metrioxenini** Voss, 1953

Subtribe **Zherichinixenina** Legalov, 2009

**Remarks**. The subtribe includes three extant genera *Lyalixena* Legalov, 2009, *Zherichinixena* Legalov, 2009, and *Prometrioxena* Voss, 1957, and two fossil genera *Paltorhynchus* Scudder, 1893 and *Succinometrioxena* Legalov, 2012.

Key to genera of Zherichinixenina

1. Sides of pronotum without teeth.............................................................*Paltorhynchus*

–Pronotum with serrated lateral carinae, sometimes teeth very weak.........................2

2. Apices of elytra with teeth...................................................................*Succinometrioxena*

–Apices of elytra without teeth...........................................................................................3

3. Femora with denticles.................................................................................*Prometrioxena*

–Femora simple.....................................................................................................................4

4. Pronotum sides almost parallel, with weak teeth. Body more elongated and flattened....................................................................................................................*Zherichinixena*

–Pronotum sides narrowed to apex, with sharp teeth. Body shorter and convex ...........................................................................................................................................*Lyalixena*

Genus *Succinometrioxena* Legalov, 2012

Type species: *Succinometrioxena poinari* Legalov, 2012 by original designation.

**Diagnosis**. Body more or less convex, black, with semierect or adpressed pale setae; rostrum subcylindrical, straight or weakly curved, slightly longer than pronotum; forehead wide, with horn-like tubercles on either side or without them; eyes finely punctate, convex and rounded; temples quite short; antennae inserted near rostrum base ventrally; antennal club two-segmented, fused, two or more times as long as eighth antennomere; pronotum bell-shaped, weakly flattened, with two longitudinal carinae and distinct pronotal constriction; lateral carinae consist of six obtuse convexities; scutellum small; elytra elongate, with two long and two short carinae; scutellar striole indistinct; striae usually irregular; interstriae weakly convex or flat; epipleuron distinct; apex of elytra separately rounded; precoxal portion of prosternum elongate; procoxal cavities separate; postcoxal portion of prosternum short; mesocoxal cavities widely separate; metepisternum narrow; abdomen convex; first and second ventrites weakly elongate; legs long; femora weakly clavate; tibiae almost straight; second and third tarsomeres bilobed; claws large, without teeth.

***Succinometrioxena andrushchenkoi*** Legalov, **sp. n.** (Figure 1).

LSIDurn:lsid:zoobank.org:act: A7B29FEA-36FB-4DB2-9C30-47A1D8C9DE54.

**Description**. Size. The body length (without rostrum), is 3.6 mm, rostrum length, and 1.1 mm. The body is black, with short, sparse, adpressed setae, appearing silvery-shiny from the presence of a cavity between the specimen and the internal surface of its impression. Rostrum is subcylindrical, quite long, about 1.1 times as long as the pronotum, 7.7 times as long as wide at the apex and in middle, 5.9 times as long as wide at the base, weakly curved, and finely punctate. Head about 0.2 times as long as the rostrum. Forehead wide, about 1.1 times as wide as rostrum base width, depressed, coarsely punctate, with horn-like tubercles on either side. Eyes quite small, rounded, distinctly convex, finely faceted. Vertex is weakly flattened, and densely punctate. Temples about 1.6 times as long as the eye, punctate. Antennae inserted near base of rostrum ventrally, under small convexities. Antennae long, almost reaching the base of the pronotum. The scape and second antennomere are almost teardrop-shaped. Scape 1.6 times as long as wide at apex. The second antennomere is 1.7 times as long as it is wide at the apex, slightly shorter than and about 0.9 times as narrow as the scape. The third–ninth antennomeres are subconical. The third–seventh antennomeres are subequal in width. Third antennomere is about 1.8 times as long as it is wide at the apex, slightly shorter, and about 0.8 times as narrow as the secooonnnd antennomere. The fourth antennomere is about 2.2 times as long as it is wide at the apex, slightly longer than the third antennomere. The fifth antennomere is about 1.6 times as long as it is wide at the apex, about 0.8 times as long as the fourth antennomere. The sixth and seventh antennomeres are subequal to the fifth antennomere. The eighth antennomere is about 1.1 times as long as it is wide at the apex, slightly shorter, and about 1.3 times as wide as the seventh antennomere. The ninth antennomere (first club article) is similar to the eighth antennomere (last article of the flagellum), about 1.6 times as long as it is wide at the apex, about 1.4 times as long, as subequal in width to eight antennomere. The tenth and eleventh antennomeres (second and third club articles) are fused, about 2.0 times as long as the ninth antennomeres. The tenth antennomere is about 1.1 times as long as wide at the apex, about 1.1 times as long, as and about 1.6 times as wide as the ninth antennomere. The eleventh antennomere is slightly shorter than its length, about 0.7 times as long, as and slightly narrower than the tenth antennomere, and weakly acuminate. The pronotum is almost bell-shaped, with a wide apical constriction, 1.4 times as long as it is wide at the apex and the base, about 1.3 times as long as it is wide in the middle, and narrower than the elytral base. The disk is densely and coarsely punctate, with two longitudinal carinae. The distance between the punctures are smaller than their diameter. The sides of pronotum with carinae consisting of obtuse convexities. The scutellum is small and triangular. The elytra are elongated and distinctly convex, about 2.3 times as long as the pronotum, about 2.4 times as long as wide at the base, and at the apical fourth about 1.8 times as long as wide in middle, with two long carinae. The humeri are weakly flattened. Scutellar striole indistinct. Punctate striae regular and distinct near elytral suture, irregular and indistinct in other elytral parts. The punctures are rounded and dense. The distance between the punctures are smaller than their diameter. The interstriae are almost flat. The epipleuron is distinct. The apex of the elytra is separately rounded, obtuse, and without elongated teeth. The margin of the elytra is sharp and carinate. The prosternum is coarsely punctate. The precoxal portion of the prosternum is elongated, 2.7 times as long as the length of the procoxa. The procoxal cavity is round and narrowly separated. The postcoxal portion of the prosternum is short. The metaventrite is about 2.2 times as long as the mesocoxal length, weakly convex, and densely punctate. The abdomen is flattened. The first ventrite is weakly elongated, about 1.4 times as long as the mesocoxal length. The second ventrite is about 0.7 times as long as the first ventrite. The third ventrite is equal to the second ventrite. The fourth ventrite is slightly shorter than the third ventrite. The fifth ventrite is about 1.7 times as long as the fourth ventrite. The legs are long. The metacoxa is transverse. The femora are weakly clavate, rugose-punctate. The trochanter is triangular. The tibiae are almost straight, weakly flattened, and weakly oblique at the apex, with apical dark setose fringes. The tarsi are long, with thick and light setae dorsally. The first tarsomere is triangular. The second tarsomere is widely bilobed. The third tarsomere elongate-bilobed. The fifth tarsomere is elongated. The claws are large, contiguous, and without teeth.

**Material examined**. Holotype, female, KRAM, no. BX 12-23, Baltic amber, late Eocene.

**Derivation of name**. The species is named in honor of K.V. Andrushchenko (Kaliningrad, Russia), who provided the type specimen for description.

**Diagnosis**. The new species differs from *S*. *poinari* in the smaller body sizes (3.6 mm), the elytral punctation is larger than the distances between them, and the rostrum is weakly curved in the females. It is distinguished from *S*. *bachofeni* and *S*. *attenuata* in the forehead without horn-like tubercles on either side of the eyes.

**Remarks**. I saw a specimen (female) of the new species on eBay.

***Succinometrioxena poinari*** Legalov, 2012 (Figure 2).

urn:lsid:zoobank.org:act:56EB377E-EC3F-4DCF-94F6-C549F57FEDD3.

**Description**. Male. Size. The body length (without rostrum) is 3.9–4.1 mm, rostrum length 1.1–1.2 mm. The rostrum is weakly curved, 1.3 times as long as the pronotum, about 6.7 times as long as wide at the apex and in the middle, densely punctate, with small granulations dorsally. The forehead is wide, flattened, coarsely punctate, and with horn-like tubercles on either side. The eyes are small, rounded, and convex. The vertex is weakly flattened and coarsely and densely punctate. The temples are short, slightly shorter than the eyes, and punctate. The antennae are inserted near the base of the rostrum ventrally, under small convexities, quite long, and almost reaching the base of the pronotum. The first antennomere is almost teardrop-shaped. The second antennomere is oval. The third–eighth antennomeres are subconical. The eighth antennomere is about 1.3 times as long as it is wide at the apex. The ninth antennomere is about 1.1 times as long as it is wide at the apex, about 1.1 times as long as and about 1.3 times as wide as the eighth antennomere. The tenth and eleventh antennomeres are fused. The tenth antennomere is about 1.4 times as long as it is wide at the apex, about 1.6 times as long as, and about 1.3 times as wide as the ninth antennomere. The eleventh antennomere is about 0.7 times as long as wide at the base, about 0.4 times as long as, and about 0.8 times as narrow as the tenth antennomere. The pronotum is bell-shaped. The disk narrowed at the apex and at the base, densely punctate, and with two longitudinal carinae. The elytra are elongated and weakly flattened, 2.6 times as long as the pronotum, and with two long and two short carinae. The carinae and seam have semierect, short, brown setae. The humeri are weakly flattened. The punctate striae are irregular and indistinct. The punctures are rounded, small, and dense. The distance between punctures subequal or smaller than their diameter. The interstriae between the punctures are weakly convex. The prosternum is coarsely punctate. The precoxal portion of the prosternum and elongated, about 1.9 times as long as the procoxal length. The procoxal cavity is round and narrowly separated by the prosternal process. The postcoxal part of the prosternum is short, about 0.6 times as long as the procoxal length. The mesocoxal cavities are rounded, and widely separated. The metaventrite is about 1.7 times as long as the mesocoxal cavity length, weakly convex, and densely punctate. Mesocoxal cavities transverse, narrowly separated. The abdomen is weakly convex, and impressed in the middle. The first ventrite is about 0.8 times as long as the metacoxal length. The second–fourth ventrites are subequal in length. The second ventrite is about 0.8 times as long as the first ventrite. The fifth ventrite is about 1.1 times as long as the fourth ventrite. The legs are long and the femora are weakly clavate, and rugose-punctate. The profemur is thicker than the meso- and metafemora. The tibiae are almost straight, weakly flattened, and weakly oblique at the apex, with an apical dark setose fringe. The tarsi is long, with thick, light, erect setae dorsally. The first tarsomere is conical. The second tarsomere is widely bilobed. The third tarsomere is elongate–bilobed. The fifth tarsomere is elongated. The claws are large, contiguous, and without teeth.

**Material examined**. Holotype, female, ISEA № 2011/1, Baltic amber, late Eocene. Specimens: male, ISEA, № 2023/1, Baltic amber, late Eocene; female, ISEA, № 2016/3, Baltic amber, late Eocene; male, CAG, 4696, Baltic amber, late Eocene.

**Remarks**. This species is fairly common in Baltic amber. I have seen at least 10 specimens on eBay and online stores.

***Succinometrioxena bachofeni*** Legalov, 2013 (Figure 3).

urn:lsid:zoobank.org:act:1E147998-E92F-4926-AB54-842BBE3DB9C9.

**Material examined**. Holotype, female, ISEA, № 2012/6, Baltic amber, late Eocene.

**Remarks**. It is known from the holotype only.

***Succinometrioxena attenuata*** Legalov et Poinar, 2020

urn:lsid:zoobank.org:act:89174A19-60FC-4E7A-B35D-2FB21576D1CB.

**Material examined**. Holotype, female, PACO № 7, Baltic amber, late Eocene. 

**Remarks**. I saw a specimen of this species from a private collection in Poland.

Key to fossil species of Metrioxenini

1. The first ventrite is strongly elongated (Figure 4B,C). Body small (2.2–2.6 mm in length) (*Archimetrioxena*, Pseudometrioxenina).........................................................................2

–The first ventrite is a little longer than the second ventrite (Figure 2 and Figure 3). The body is large (3.5–8.3 mm in length) (Zherichinixenina)........................................................................3

2. The forehead has small horn-like tubercles on either side of the eyes. The pronotum is densely punctated, almost matte (Figure 5).........................................*A*. *electrica*

–The forehead is simple. The pronotum is sparsely punctated and lustrous (Figure 4)....................................................................................................................................*A*. *zherikhini*

3. The sides of the pronotum have obtuse teeth. The forehead usually has horn-like tubercles on either side (Figure 1, Figure 2 and Figure 3). (*Succinometrioxena*).......................................................4

–The sides of the pronotum have no teeth. The forehead is always without horn-like tubercles on either side (Figure 6). (*Paltorhynchus*)....................................................................7

4. The forehead has horn-like tubercles on either side of the eyes (Figure 1 and Figure 2)....5

–The forehead is without horn-like tubercles on either side of the eyes (Figure 3A)...6

5. The elytral punctation is larger than the distances between them. The rostrum is weakly curved in the females (Figure 1). The body (without rostrum) is smaller (3.6 mm).......................................................................................................................*S*. *andrushchenkoi*

–The elytral punctation is smaller than the distances between them (Figure 2). The rostrum is straight in the females. The body (without rostrum) is larger (4.1–4.3 mm)…………………………………………………………………………………….....*S*. *poinari*

6. The body (without rostrum) is smaller (3.5 mm). The elytra are wider, with small punctation.....................................................................................................................*S*. *bachofeni*

–The body (without rostrum) is larger (6.3 mm). The elytra are narrower, with large punctation......................................................................................................................*S*. *attenuata*

7. The body (without rostrum) smaller (3.7 mm) has a subdued sculpture............................................................................................................................*“P.” bisculcatus*

–The body (without rostrum) is larger (7.5–8.3 mm) with a coarse sculpture (Figure 6)........................................................................................................................................*P*. *narwhal*


**List of fossil Metrioxenini**


Pseudometrioxenina Legalov, 2023*Archimetrioxena* Voss, 1953= *Palaeometrioxena* Legalov, 2012*A. electrica* Voss, 1953—Baltic amber [22]*A*. *zherikhini* (Legalov, 2012)—Baltic amber [25]Zherichinixenina Legalov, 2009Succinometrioxena Legalov, 2012*S. andrushchenkoi* Legalov, sp. n.—Baltic amber*S. attenuata* Legalov et Poinar, 2020—Baltic amber [28]*S*. *bachofeni* Legalov, 2013—Baltic amber [27]*S*. *poinari* Legalov, 2012—Baltic amber [26]*Paltorhynchus* Scudder, 1893*P*. *narwhal* Scudder, 1893—Florissant [24] (Figure 6)*“P.” bisculcatus* Scudder, 1893—Roan Mountain [24]

**Remarks**. Scudder [24] described a smaller species (3.7 mm) conditionally placed in the genus *Paltorhynchus*. The shape of the body and the carinae on the elytra confirm the correct placement in the tribe Metrioxenini. However, the structure of the abdomen is not known, which does not allow it to be reliably placed in any subtribe. The body length of Metrioxenina is 2.1–3.8 mm, but usually does not exceed 3 mm; Zherichinixenina species range from 3.5 to 8.3 mm. Since it is habitually similar to *P*. *narwhal*, I keep it in the genus *Paltorhynchus*, but to make a conclusion about the systematic position, it is necessary to study the types or new materials from the locality.

## 4. Discussion

The family Belidae Schoenherr, 1826 is a small relict group of primitive weevils [14,34,35,36,37]. One extinct (Montsecbelinae Legalov, 2015) and two Recent subfamilies (Belinae Schoenherr, 1826 and Oxycoryninae Schoenherr, 1840) form this family [14,15,35]. The last two are quite isolated and were previously considered as two separate families [15,38,39,40]. The belinae are characterized by the body being more or less cylindrical and second tarsomere more or less elongated, not bilobed, and distributed in the southwest and east of the central part of South America, in Australia, New Guinea, Moluccas, Aru Is., Lord Howe Is., Solomon Is., and New Zealand [14,15,36,38,41,42,43]. Oxycoryninae live in Central America, the West Indies, Western and central South America, Melanesia (including Hawaii), North Africa, southern South Africa, Southeast Asia, the Sunda Islands, New Caledonia, and New Zealand [13,14,15,20,32,36,43,44,45,46,47,48]. The first record of belids is the Montsecbelinae from the early Cretaceous of Spain and France [35,49,50]. The beetle described in the family Belidae from the middle Jurassic of China [51] turned out to belong to the tribe Probelini Legalov, 2009 of the subfamily Eobelinae Arnoldi, 1977 of the family Nemonichidae Bedel, 1882, rich in the Jurassic of Kazakhstan [52]. Belinae are found in the early Cretaceous of Brazil, where the extinct tribe Davidibelini Legalov, 2015 is represented [35,53]. The modern tribes Belini Schoenherr, 1826, Agnesiotidini Zimmerman, 1994 and Pachyurini Kuschel, 1959 are not known in the fossil state [14,35]. The oldest Oxycoryninae (*Khetana decapitata* Zherikhin, 1993) were found in the Albian of the Russian Far East [35,54]. The tribe Burmocorynini Legalov, 2020 described from mid-Cretaceous Burmese [55] amber has been placed in Mesophyletinae Poinar, 2006 [56] of the family Ithyceridae Schoenherr, 1823. The first Paleogene Oxycoryninae was found in the middle Eocene of the USA [24]. The tribes Metrioxenini and Palaeorhopalotriini Legalov, 2013 are found in the late Eocene of Europe and the terminal Eocene of the USA [24,27]. *Pleurambus strongylus* Poinar et Legalov, 2014 from the modern tribe Allocorynini Sharp, 1890 was described from early Miocene Dominican amber [57]. There are no other fossil finds of Belidae.

The weevils of the tribe Metrioxenini are confined to the palms of the genus *Arenga* [13]. Their larvae develop in the stems of these plants [13], but it is possible that in some genera they develop in fruits. They have not been found on other genera of the tribe Caryoteae, but they may be covered with them.

Palms of the genus *Arenga* include more than 20 species distributed (Figure 7) in southern China and South and Southeast Asia (including the Philippines, Sunda, New Guinea, and Ryukyu), and northern Australia [58]. The tribe Caryoteae Scheff. also includes the genera *Caryota* L. and *Wallichia* Roxb., distributed in the Oriental region.

The current distribution of the tribe Metrioxenini covers Southeast Asia, from China (Yunnan), Indochina (Vietnam, Thailand, Laos, Malaysia), the Sunda Islands (Java, Sumatra, Kalimantan, Sulawesi, Timor, etc.), and the Philippines (Figure 7). Thus, the area of the tribe Metrioxenini completely fits the area of the genus *Arenga*. It can be assumed that the Metrioxenini species will be found in most of the range of the *Arenga* palm.

The earliest finds of the subfamily Coryphoideae Burnett are from the Aptian of North Africa [59]. Reliable macrofossils of representatives of the genus *Arenga* are not known [60]. Seed from the middle Oligocene of Puerto Rico was considered to resemble closely those of the genera *Arenga* or *Iriartea* Ruiz & Pav. [61]. The earliest fossil pollen of *Arenga* is from the early Eocene of India [62] and the early Miocene of Kalimantan [63]. Seeds probably corresponding to the genus *Caryota* were described from the early Eocene London Clay [64]. Pollen similar to *Caryota* was recorded from the Oligocene of the Isle of Wight [65]. Three palm species (*Phoenix eichleri* Conwentz, 1886, *Palmophyllum kuenowi* (Casp. 1872), *Bembergia pentatrias* Caspary 1881) are known from Baltic amber [2]. *Phoenix eichleri* belongs to the subfamily Coryphoideae of the tribe Phoeniceae. *Palmophyllum kuenowi* and *Bembergia pentatrias* have an unclear taxonomic position. Palms are known from the Green River, including the modern genus *Phoenix* L. [66]. Indeterminate retirements of palm trees are found in Florissant [67].

Fossil records of Metrioxenini are from the early Eocene of Roan Mountain, Colorado (USA), the late Eocene of Europe (Baltic amber), and the terminal Eocene of Florissant, Colorado (USA) (Figure 7). These finds show that in the Eocene the tribe was distributed much wider than its modern range. The question arises of what plants it was associated. Since modern representatives of Metrioxenini are monophagous on the *Arenga* palms, it can be assumed that fossil representatives developed on the same food plant or genera close to it. 

## 5. Conclusions

It can be assumed that the range of the genus *Arenga* in the Eocene was very wide and its species overlapped with Metrioxenini weevil localities. Hopefully, the palm species can be identified during further paleobotanical research.

## Figures and Tables

**Figure 1 life-13-01121-f001:**
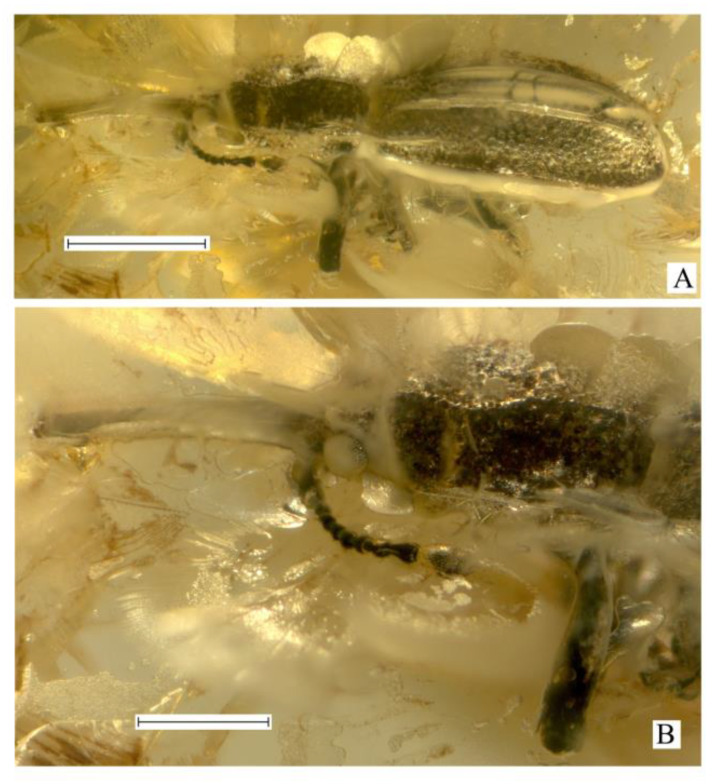
*Succinometrioxena andrushchenkoi* n. sp., female, holotype, KRAM, no. BX 12-23: (**A**) habitus, lateral view; (**B**) rostrum and head, dorso-lateral view. Scale bar 1.0 mm for (**A**), 0.5 mm for (**B**).

**Figure 2 life-13-01121-f002:**
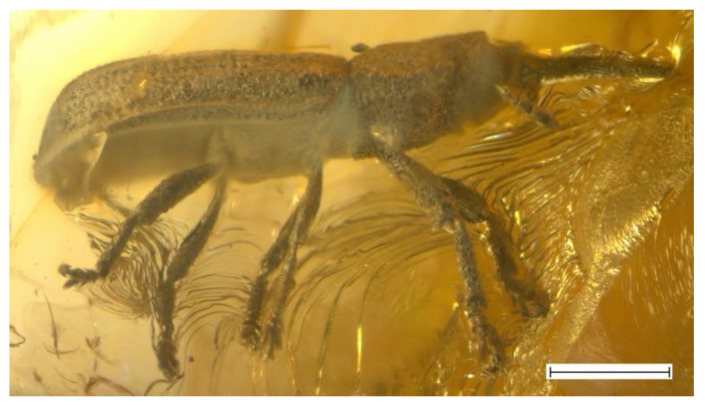
*Succinometrioxena poinari*, male, specimen, ISEA, № 2023/1, habitus, lateral view. Scale bar 1.0 mm.

**Figure 3 life-13-01121-f003:**
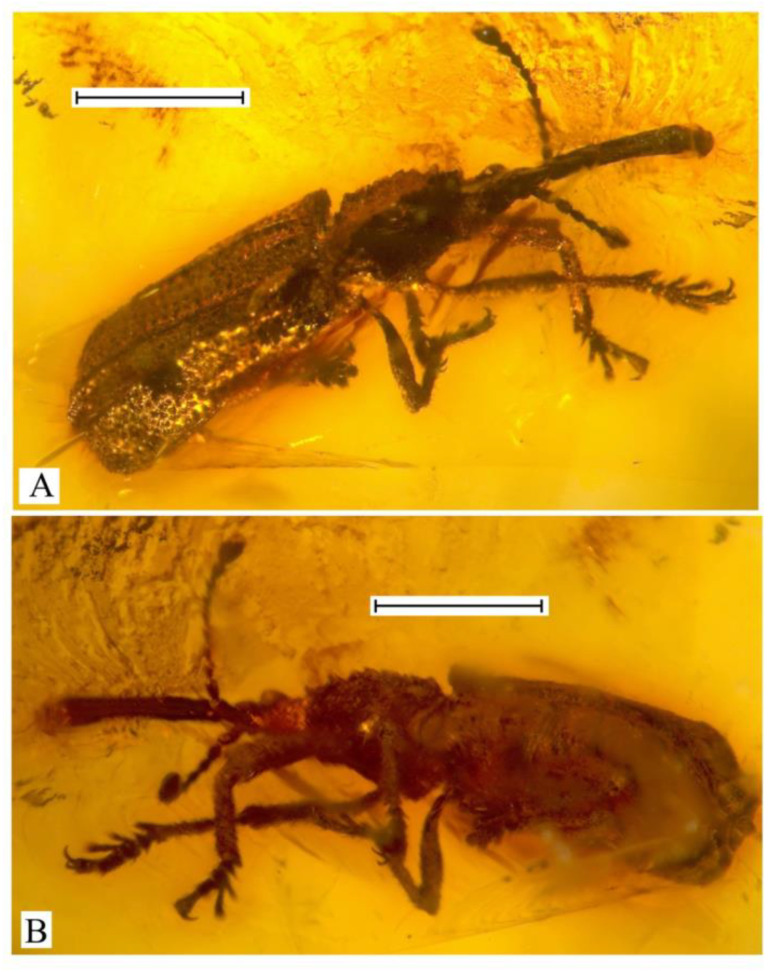
*Succinometrioxena bachofeni*, female, holotype, ISEA, № BA2012/6, habitus: (**A**) dorso-lateral view; (**B**) ventral view. Scale bar 1.0 mm.

**Figure 4 life-13-01121-f004:**
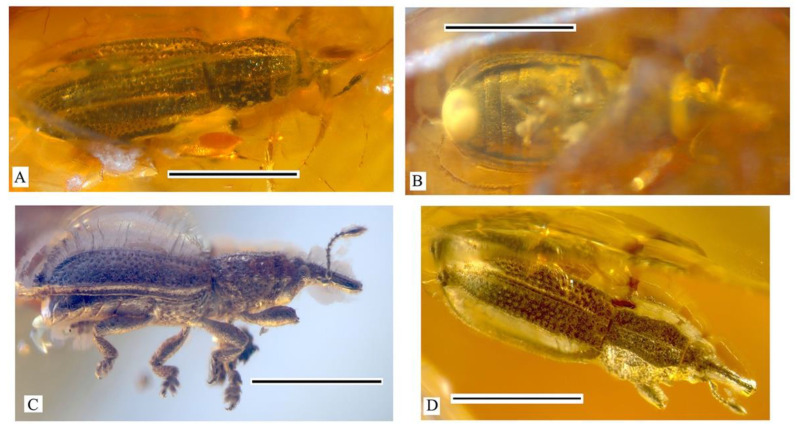
*Archimetrioxena zherikhini*, habitus: (**A**) female, holotype, PIN № 964/1236, dorsal view; (**B**) female, holotype, PIN № 964/1236, ventral view, **left**; (**C**) male, specimen, ISEA № BA2012/18, lateral view, **right**; (**D**) male, specimen, ISEA № BA2012/18, dorsal view. Scale bar 1.0 mm.

**Figure 5 life-13-01121-f005:**
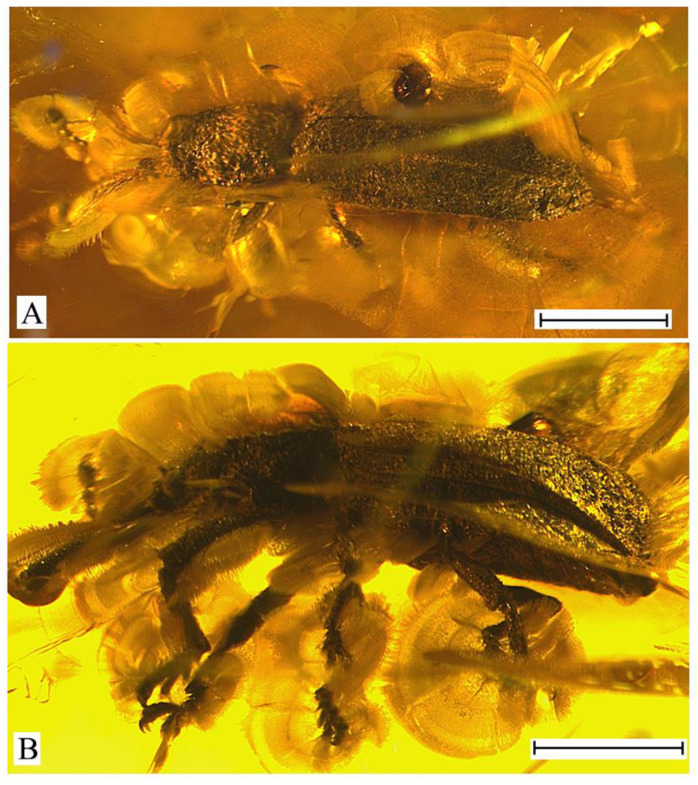
*Archimetrioxena electrica*, female, holotype, GPIH № 194, habitus: (**A**) dorsal view; (**B**) lateral view, left. Scale bar 1.0 mm.

**Figure 6 life-13-01121-f006:**
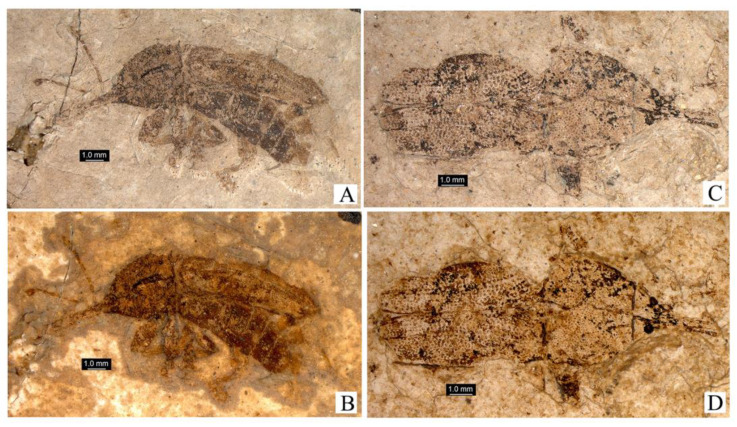
*Paltorhynchus narwhal*, habitus: (**A**) female, lectotype, MCZ № 12247, lateral view, **left**; (**B**) female, lectotype, MCZ № 12247, lateral view, **left**, with alcohol; (**C**) male, paralectotype, MCZ № 463, dorsal view; (**D**) male, paralectotype, MCZ № 463, dorsal view, with alcohol.

**Figure 7 life-13-01121-f007:**
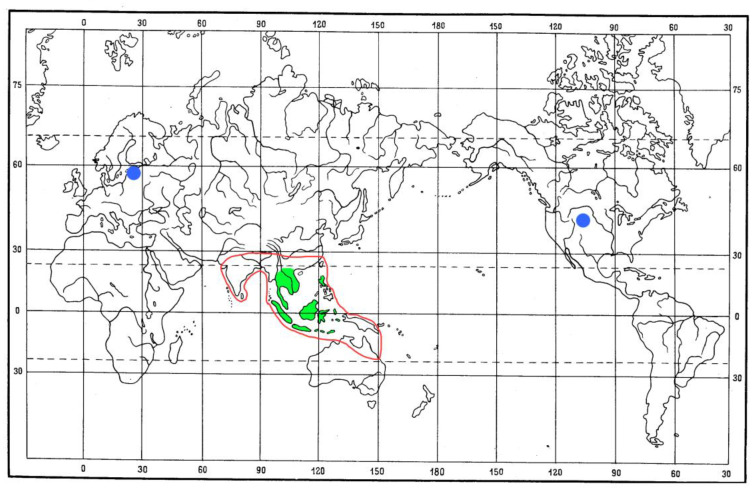
Distribution of the species of the tribe Metrioxenini and genus *Arenga*: Recent members of Metrioxenini—green shaded area, red line—distribution of *Arenga*; blue circle—fossil record of Metrioxenini.

## Data Availability

The specimens are deposited in the Kaliningrad Regional Amber Museum, Kaliningrad, Russia and the Institute of Systematics and Ecology of Animals, Siberian Branch, Russian Academy of Sciences, Novosibirsk.

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
