# Peer review of "Were Arenga Palms (Arecaceae) Present in the Eocene? A Review of the Genus Succinometrioxena Legalov, 2012"

_life, 2023, doi:10.3390/life13051121_

Round 1

Reviewer 1 Report

See the attached file with suggestions and minor corrections

Author Response

I thank the reviewer for the comments.

what does it means? Did the author actually studied this example?

-This means that the author saw a photograph of this species, which showed diagnostic signs.

again, what does it means? How the author can be sure that these fossil specimens do actually belong th S. poinari?

-

I am sure of their species placement, since the photographs were in high quality and signs allowing them to be attributed to S. poinari were visible.

ADJUST THE DICHOTOMOUS KEY MARGINS

-It was done

Reviewer 2 Report

I suggest a slight correction in a few places of the text as noted in the manuscript - attached file.

Author Response

I thank the reviewer for the comments. I corrected the ms.

Reviewer 3 Report

The topic of the MS “Were Arenga palms present in the Eocene? A review of the genus Succinometrioxena Legalov, 2012” is relevant to the journal “MDPI Life”. The paper is original (not published as far I know) and provide new data concerning fossil Metrioxenini weevils and the possible fodder plant distribution of these beetles. The paper adds to science one new Succinometrioxena species from Baltic amber. The paper is useful and well-imaged contribution in the subject area. The conclusions in the paper and placement of the specimen into the genus are ok and justified. The paper is scientifically correct.  References are appropriate. The existing list of references is ok, and I don’t see any excess citations. The idea about trophic relationships and reconstruction of the distribution of host plants in the past with use of fossil, probably monophagous, beetles is interesting and justified.

My possible recommendation for the MS is following:

1. Mention the family name of the studied beetles (Belidae) and Arenga palms (Arecales: Arecaceae) in summary or in key words.

2. Add remarks to Succinometrioxena bachofeni. Maybe “known from the holotype only”, if so.

3. Add collection number for the holotype of the new species Succinometrioxena andrushchenkoi. The holotype is deposited in KRAM, but the collection number is not mentioned in the text. The short description of the amber piece (e.g. sizes, syninclusions) with this inclusion is also recommended.

4. Line 383: change “tribes Caryoteae” into “tribe Caryoteae”. 

5. Lines 413-414: delete the phrase “Can you say more about Arenga palms (fan or pinnate leaved, height, etc.) and perhaps show a photo of one of the fossils, or a recent species.” It is a comment or a part of dialogue.

6. Line 417-418: the phrase “Thus, in weevil localities, while the remains of palms were found, they could not be identified to genus or tribe.” is a little strange for conclusions. Maybe “thus” should be deleted, I don’t known. So, I recommend to check the phrase, maybe, add or delete something. “Conclusions” is separate chapter, it should be clearly formulated.

7. The goals of provided description of Succinometrioxena poinari Legalov, 2012 is not clear. Is it re-description of the species and some emendations are mentioned here? Several additions and details are added to the species characteristic, are not it? I suppose, it should be mentioned in the summary or in the text (e.g. additional specimens of Succinometrioxena poinari Legalov, 2012 are studied …”). Or, maybe, this part can be reduced (the new species is diagnosed, all species are keyed, other fossil Metrioxenini are not so detailed presented). Is such long descriptive part for described in 2012 year species useful? Why?

8. Figure 1 in lines 122-124. Figure 1A (lateral view) and Figure 1B (dorso-lateral view) are identical and represent the specimen in the dorso-lateral view.  I recommend: (a) delete one figure or (b) provide an additional figure of lateral view.

9. Maybe, the phrase from lines 15-17 “It has been suggested that palms of the genus Arenga or forms close to it were distributed in the Eocene of North America and Europe. Records of Metrioxenini which are monophages on these palms confirm this assumption.” could be transferred into final part of summary. It would be more logical, to unite the text parts on palms and distribution of palms together. It is only a recommendation. 

So, in sum, the MS is clear and interesting, it is good and can be published after minor corrections.

Author Response

I thank the reviewer for the comments.

1. Mention the family name of the studied beetles (Belidae) and Arenga palms (Arecales: Arecaceae) in summary or in key words.

- It was done.

2. Add remarks to Succinometrioxena bachofeni. Maybe “known from the holotype only”, if so.

- It was done.

3. Add collection number for the holotype of the new species Succinometrioxena andrushchenkoi. The holotype is deposited in KRAM, but the collection number is not mentioned in the text. The short description of the amber piece (e.g. sizes, syninclusions) with this inclusion is also recommended.

4. Line 383: change “tribes Caryoteae” into “tribe Caryoteae”. 

- It was done.

5. Lines 413-414: delete the phrase “Can you say more about Arenga palms (fan or pinnate leaved, height, etc.) and perhaps show a photo of one of the fossils, or a recent species.” It is a comment or a part of dialogue.

- It was done.

6. Line 417-418: the phrase “Thus, in weevil localities, while the remains of palms were found, they could not be identified to genus or tribe.” is a little strange for conclusions. Maybe “thus” should be deleted, I don’t known. So, I recommend to check the phrase, maybe, add or delete something. “Conclusions” is separate chapter, it should be clearly formulated.

- It was corrected.

7. The goals of provided description of Succinometrioxena poinari Legalov, 2012 is not clear. Is it re-description of the species and some emendations are mentioned here? Several additions and details are added to the species characteristic, are not it? I suppose, it should be mentioned in the summary or in the text (e.g. additional specimens of Succinometrioxena poinari Legalov, 2012 are studied …”). Or, maybe, this part can be reduced (the new species is diagnosed, all species are keyed, other fossil Metrioxenini are not so detailed presented). Is such long descriptive part for described in 2012 year species useful? Why?

- There was no description of the male, and I give him the first. Female's description was deleted.

  1. Figure 1 in lines 122-124. Figure 1A (lateral view) and Figure 1B (dorso-lateral view) are identical and represent the specimen in the dorso-lateral view.  I recommend: (a) delete one figure or (b) provide an additional figure of lateral view.

- It was done.

  1. Maybe, the phrase from lines 15-17 “It has been suggested that palms of the genus Arengaor forms close to it were distributed in the Eocene of North America and Europe. Records of Metrioxenini which are monophages on these palms confirm this assumption.” could be transferred into final part of summary. It would be more logical, to unite the text parts on palms and distribution of palms together. It is only a recommendation. 

- I'd rather leave things as they are.
